

# Comparison of the elastic recovery and strain-in-compression of commercial and novel vinyl polysiloxane impression materials incorporating a novel crosslinking agent and a surfactant

Shahab Ud Din[1], Osama Khattak[2], Farooq Ahmad Chaudhary[1], Asfia Saeed[3], Azhar Iqbal[2], Jamaluddin Syed[4], Alaa Ahmed Kensara[5], Thani Alsharari[6], Mohammed Mustafa[7], Sherif Elsayed Sultan[8] and Mangala Patel[9]

[1] School of Dentistry (SOD), Shaheed Zulfiqar Ali Bhutto Medical University (SZABMU), Islamabad, Pakistan
[2] Department of Restorative Dentistry, Jouf University, Sakaka, Saudi Arabia, Sakaka, Saudi Arabia
[3] Shifa College of Dentistry, Shifa Tameer-e-Millat University, Islamabad, Pakistan
[4] Oral Basic Clinical Sciences Department, Faculty of Dentistry, King AbdulAziz University, Jeddah, Saudi Arabia
[5] Department of Oral and Maxillofacial Prosthodontics, King Abdul Aziz University Dental Hospital, Jeddah, Saudi Arabia
[6] Department of Restorative and Dental Science, Faculty of Dentistry, Taif University, Taif, Saudi Arabia
[7] Department of Conservative Dental Sciences, College of Dentistry, Prince Sattam Bin Abdulaziz University, Al-Kharj, Saudi Arabia
[8] Department of Fixed Prosthodontics, Tanta University, Tanta, Egypt
[9] Centre for Oral Bioengineering (Dental Physical Sciences Unit), Bart's and The London, School of Medicine and Dentistry, Queen Mary University of London, London, UK

Corresponding authors
Osama Khattak,
dr.osama.khattak@jodent.org
Farooq Ahmad Chaudhary,
chaudhary4@hotmail.com

## ABSTRACT

This study aims to formulate experimental vinylpolysiloxane (VPS) impression materials and compare their elastic recovery and strain-in-compressions with three commercial VPS materials (Aquasil, Elite, and Extrude). Five experimental materials (Exp), two hydrophobic (Exp-I and II) and three hydrophilic (Exp-III, IV and V) were developed. Exp 1 contained vinyl-terminated poly-dimethyl siloxane and a conventional crosslinking agent (poly methylhydrosiloxane), while Exp- II contained a novel crosslinking agent that is tetra-functional dimethyl-silyl-ortho-silicate (TFDMSOS). Exp III–V (hydrophilic materials) were formulated by incorporating different concentrations of non-ionic surfactant (Rhodasurf CET-2) into Exp II formulation. Measurement of elastic recovery and strain-in-compression for commercial and experimental materials were performed according to ISO4823 standard using the calibrated mechanical testing machine (Tinius Olsen). One-way analysis of variance (one-way ANOVA) and Tukey's *post-hoc* (HSD) test were used for statistical analysis and a *p*-value of $\leq 0.05$ was considered significant. Exp-I has statistically similar values to commercial VPS. The Exp-II showed the highest elastic recovery, while % elastic recovery was reduced with the addition of the non-ionic surfactant (Rhodasurf CET-2). The % reduction was directly related to the concentration of Rhodasurf CET-2. In addition, Exp II had significantly higher strain-in-compression values compared to Exp-I and commercial materials. These values were further increased with the addition of a non-ionic surfactant (Rhodasurf CET-2) was added (Exp-III, IV and V).

## INTRODUCTION

Dental impression refers to a negative imprint of oral hard and soft tissues (*Balkenhol et al., 2010*). An accurate impression is of utmost importance for the successful fabrication of a prosthesis. Ideally, the impression material should have good wettability, accuracy, elasticity and minimal distortion on removal and storage (*Balkenhol et al., 2010*). Impression materials are compressed against the tray, especially while recording undercut areas, and on the removal of impression from the mouth. The degree of distortion of the material depends on the severity of the undercut, elastic recovery of the material, the time the material is kept in the compressed state and storage conditions (*Balkenhol et al., 2010*; *Blomberg et al., 1992*).

The elastic recovery of impression material is the capacity of the material to revert to its original position, without significant distortion after being strained, when the deforming force is removed (*Hamalian, Nasr & Chidiac, 2011*). It is due to the presence of folded polymeric segments within the material, which coil and uncoil during loading and unloading. Therefore, the greater the elastic recovery of the material, the more precise the prosthesis.

The likelihood of permanent deformation increases on slow removal of an impression as the material is stressed for longer duration (*Balkenhol et al., 2010*; *Din et al., 2018b*; *Din et al., 2021*; *Din et al., 2022*; *Hondrum, 1994*; *Mandikos, 1998*). None of the impression materials has 100% elastic recovery (*Hamalian, Nasr & Chidiac, 2011*); rather, most elastomeric materials exhibit time-dependent recovery from deformation (viscoelasticity) (*Braden et al., 1997*; *Darvell, 2009*; *Goldberg, 1974*). The elastic recovery of these materials depends on their composition, such as the pre-polymer, cross-linking agents, and fillers (*Deb, 1998*; *Din et al., 2018a*; *Din et al., 2018b*; *Klooster, Logan & Tjan, 1991*; *Lawson, Burgess & Litaker, 2008*; *Saeed et al., 2022*; *Ud Din et al., 2018*).

The International Standards Organisation (*ISO4823, 2007*) recommends that an elastomeric impression material (all consistencies) must have 96.5% elastic recovery. Although all elastomeric impression materials fulfil the criteria, the VPS possesses better elastic recovery compared to other impression materials (*Bonsor & Pearson, 2013*; *Din et al., 2018b*; *Hamalian, Nasr & Chidiac, 2011*; *Klooster, Logan & Tjan, 1991*). This allows pouring of the impression to fabricate cast after six minutes of removal from the mouth (*Blomberg et al., 1992*). In addition, these materials exhibit great dimensional stability and high tear strength.

Different brands of VPS impression materials showed variations in elastic recovery. *Lawson, Burgess & Litaker (2008)* investigated the elastic recovery for five VPS and a hybrid impression material (containing siloxane and polyether groups) after subjecting materials to tensile and compressive stress. The VPS impression materials showed improved elastic recovery in comparison to the hybrid material, which may be related to the compositions

of materials as hybrid material composed of polyethers, which have a lower elastic recovery compared to VPS (*Hondrum, 1994*; *Ud Din et al., 2022a*; *Ud Din et al., 2022b*).

Strain-in-compression is the measurement of the stiffness or flexibility of impression material. It determines the ability of polymerized material to be removed from the mouth or cast without permanent deformation, injury to oral tissues or fracture. Also, it dictates the ability of the impression to resist deformation and withstand the weight of the dental stone when the cast is poured (*Helvey, 2011*; *Klooster, Logan & Tjan, 1991*; *Lu, Nguyen & Powers, 2004a*).

To overcome the problem of inherent hydrophobicity of VPS and to improve tear strength and % elongation at break of the material, in our previous work, novel formulations of VPS were fabricated using a novel cross-linking agent *i.e.,* tetra-functional (dimethylsilyl) ortho-silicate (TFDMSOS) and novel surfactant *i.e.,* Rhodasurf CET-2 (ethoxylated cetyloleyl alcohol (*Din et al., 2018a*; *Din et al., 2018b*; *Din et al., 2017*). The addition of TFDMOS improved mechanical properties of experimental impression materials, while the non-ionic surfactant was added to improve wetting properties of the materials. Different researchers have explored the effect of various surfactants to improve the hydrophilicity of the material, however, little work has been carried out to improve the tear strength of VPS impressions. Additionally, the effects of the addition of surfactant on the mechanical properties of the materials and the hydrophilicity of these modified materials after disinfection requires further exploration. Rhodasurf CET-2 is a non-ionic surfactant which is made by combination of ethoxylated cetyl and ethoxylated oleyl alcohols. Ethoxylated oleyl alcohol has a double bond in its chemical structure. The double bond could be possibly activated during mixing and played a vital role in the cross-linking polymerization reaction leading to improved elastic recovery and strain-in-compression (*Singer et al., 2022*).

*Ud Din et al. (2018)* observed that the incorporation of a novel cross-linking agent (TFDMSOS) significantly improved the materials' % elongation-at-break and tear strength compared to the control containing a conventional crosslinking agent-poly (methyl-hydro siloxane). Additionally, the incorporation of a novel surfactant (Rhodasurf CET-2) further resulted in a significant increase in % elongation-at-break (*Din et al., 2018a*). It was also noted that the experimental formulation has a lower contact angle (improved hydrophilicity) than commercial formulations. Additionally, disinfection has little impact on the contact angle as the surfactant did not readily leach out in a disinfecting solution (*Din et al., 2017*). However, mechanical testing including elastic recovery and strain-in-compression required further exploration before considering the experimental formulation as a better alternative to commercially available VPS impression materials.

The purpose of this study was to evaluate the effect of a novel cross-linking agent, TFDMSOS and novel surfactant (Rhodasurf CET-2) on the elastic recovery and strain-in-compression of experimental VPS and to compare it with commercial materials. In summary, addition silicone materials with higher cross-link density have better elastic recovery, as they have a greater number of cross-links that can resist deformation. This property, along with other desirable characteristics such as tear strength and dimensional stability, make addition silicone impression materials a popular choice

**Table 1** Composition of novel experimental (Exp-I, II, III, IV and V) VPS impression materials.

| Components | Exp-I | Exp-II | Exp-III | Exp-IV | Exp-V |
|---|---|---|---|---|---|
| | Base Paste (Wt %) | | | | |
| Vinyl-terminated dimethylpolysiloxane (Mw 62700) | 39.90 | 39.90 | 37.95 | 37.46 | 36.98 |
| Polymethylhydrosiloxane (Mw 2270) | 1.10 | 0.77 | 0.74 | 0.73 | 0.72 |
| Tetra-functional (dimethylsilyl) orthosilicate (TFDMSOS) (Mw 329) | – | 0.33 | 0.32 | 0.31 | 0.31 |
| Filler Aerosil R 812 | 9 | 9 | 9 | 9 | 9 |
| **Components** | Catalyst Paste (Wt %) | | | | |
| Vinyl-terminated dimethylpolysiloxane (Mw 62700) | 40.72 | 40.72 | 39.51 | 39.51 | 39.51 |
| Platinum (0.05 M) | 0.06 | 0.06 | 1.27 | 1.27 | 1.27 |
| Palladium ($<1\mu$m) | 0.23 | 0.23 | 0.22 | 0.22 | 0.22 |
| Filler Aerosil R 812 | 9 | 9 | 9 | 9 | 9 |
| Rhodasurf CET-2 (non-ionic surfactant) | – | – | 2.00 | 2.50 | 3.00 |

for dental impressions. It was hypothesized that in the current study the experimental formulations have better elastic recovery and strain-in compression-values due to higher cross-links provided by adding TFDMSOS and Rhodasurf CET-2 and making it a more suitable material for recording an accurate impression.

## MATERIALS & METHODS

The ethical approval and informed consent were not required for this study, since this study do not involve living human subjects and only involve *in vitro* laboratory testing of dental impression material. Three medium-body commercial VPS impression materials were used in this study; Aquasil Ultra Monophase, USA, Dentsply (Aq M), Elite HD Monophase, Italy, Zhermack (Elt M) and Extrude, USA, Kerr (Extr M). Additionally, five experimental VPS formulations were prepared as base paste and catalyst paste (Table 1) *Darvell, 2009*. Exp-I was used as a control for Exp-II, while Exp-II acted as a control group for Exp-III, Exp-IV and Exp-V.

### Preparation of experimental formulations

The base paste of Exp-1 (hydrophobic VPS) was formulated by mixing vinyl-terminated poly-dimethyl siloxane and a conventional cross-linking agent (poly methylhydrosiloxane) for 5 min using an electric hand mixer (Kenwood, kMix, UK). The filler (Aerosil R812S) was added to the mixture and a uniform paste was made by mixing the components with a pestle and mortar for 5 min, followed by blending the paste with an electric mixer for 10 min.

The catalyst paste was formulated by mixing vinyl-terminated poly (dimethylsiloxane), platinum catalyst and palladium for 5 min with the electric hand mixer, followed by the addition of filler (Aerosil R812S) and mixing it with the pestle and mortar and electric hand mixer. For preparation of Exp-II impression material, the amount of poly (methylhydrosiloxane) was reduced from 1.10% to 0.77% and it was replaced it with a novel cross-linking agent (TFDMSOS) in the base paste of Exp-I formulation. Vinyl to

silane groups were maintained at a 1:1 ratio. The catalyst paste for Exp-II was similar to that of Exp-I.

Experimental formulations III, IV and V were formulated by modifying base-paste of Exp-II with addition of non-ionic surfactant; Rhodasurf CET-2) at concentrations of 2%, 2.5% and 3% respectively. The quantities of constituents in the catalyst paste were adjusted to ensure adequate polymerization of the materials (Table 1). The prepared base and catalyst paste of experimental materials were kept in separate compartments of cartridge and stored at 4 °C $\pm$ 2 °C.

## Sample preparation for elastic recovery and strain under compression

The cylindrical samples for elastic recovery and strain-in-compression were prepared using polytetrafluoroethylene (PTFE) mould measuring 20 mm in length $\times$ 12.5 mm in diameter according to ISO4823 (2007) standard. PTFE mould was positioned on top of a metal plate lined with an acetate sheet. The base and catalyst pastes were syringed into the mould using an auto-mixing syringe and the mould was sandwiched by another acetate-lined metal plate. The assembly was held using C-clamp. Commercial materials were left to polymerize according to the manufacturer's instructions while experimental materials were allowed to be set for 4 to 11 min (Din et al., 2017).

To measure elastic recovery ($n = 12$), two metal plates ($13 \times 13 \times 3$ mm$^3$) were fixed on either side of the specimen with the aid of double-sided sticky tape. The length of the specimen including metal plates ($h_1$) was recorded using a digital micrometre (Mitutoyo, Japan) to an accuracy of 0.001 mm. Then the specimen was deformed to $6 \pm 0.1$ mm within 1 s using the calibrated mechanical testing machine (Model H5KS, load cell 5kN; Tinius Olsen, Ltd., Salfords, UK) shown in Fig. 1. The deformation force was released slowly over a period of 5 s. After two minutes the length was measured again ($h_2$). The elastic recovery in percentage, K, was assessed using Eq. (1).

$$\mathbf{K} = 100 - \left[ 100 \left( \frac{\mathbf{h_1 - h_2}}{\mathbf{ho}} \right) \right] \tag{1}$$

$h_0$ is the height (mm) of the split mould
$h_1$ is the length (mm) of the specimen immediately before the application of the initial load
$h_2$ is the length of the specimen, 2 min after removing the deformation force

To evaluate strain-in-compression, 12 samples per material were tested. An initial force of $1.22 \pm 0.1$ N was exerted on the specimen and the distance ($h_1$) was calculated using the Tinius Olsen (Fig. 1). The load was increased to $12.25 \pm 0.1$ (N) progressively over a time of 10 s at a rate of three mm/min and a change in height of the specimen was noted again ($h_2$). The percentage of strain-in-compression, E, was calculated using Eq. (2).

$$\mathbf{E} = \left( \frac{\mathbf{h_1 - h_2}}{\mathbf{h_0}} \right) 100 \tag{2}$$

$h_0$ is the height (mm) of the split mould
$h_1$ is the length (mm) of the specimen, 30 s after submission of the opening load
$h_2$ is the length of the specimen, 30 s after submission of the amplified load.

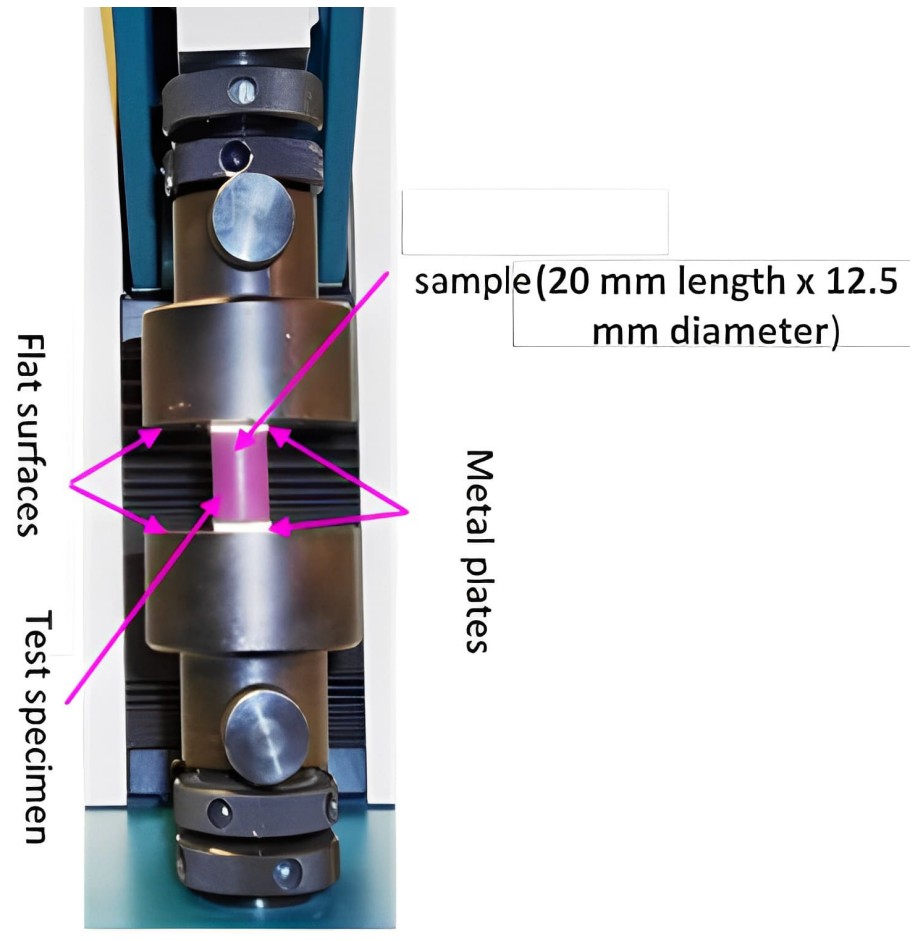

**Figure 1** Tinnus Oslen assembly used to determine elastic recovery and strain-in-compression of experimental and commercial polymeric vinylpoly siloxane impression materials.

The data was analyzed using SPSS Version 22 (Armonk NY IBM Corp, Armonk, NY, USA). Numerical data were presented as mean and standard deviation. Analysis of variance was performed with *p* value at 0.05. Where significant difference in group was found, individual means were compared using *post-hoc* Tukey's test.

## RESULTS

Table 2 shows the elastic recovery and strain-in-compression for commercial and experimental VPS impression materials immediately after setting. All the tested materials met the ISO4823 requirement of having elastic recovery greater than 96.5%. Exp-II exhibited the highest elastic recovery while Exp-V demonstrated the lowest values. The *post-hoc* analysis revealed that all three commercial products and Exp-I had statistically similar elastic recoveries.

Use of novel crosslinking agent (TFDMSOS) instead of conventional agent significantly increased elastic recovery. A significant difference in the elastic recovery was noted between

**Table 2   Average elastic recovery and strain in compression of commercial and experimental VPS immediately after setting.**

| Impression materials | Elastic recovery (%) | Strain-in-compression |
|---|---|---|
| Aq M | 99.32 ± 0.30 | 4.261 ± 0.154 |
| Elt M | 99.31 ± 0.35 | 3.153 ± 0.177 |
| Extr M | 99.27 ± 0.32 | 4.405 ± 0.118 |
| Exp-I | 99.32 ± 0.52 | 4.677 ± 0.207 |
| Exp-II | 99.65 ± 0.09 | 5.360 ± 0.163 |
| Exp-III | 99.50 ± 0.23 | 6.137 ± 0.256 |
| Exp-IV | 99.37 ± 0.26 | 6.541 ± 0.239 |
| Exp-V | 99.12 ± 0.16 | 7.076 ± 0.220 |

Exp-II and Exp-V. It was noted that the addition of a non-ionic surfactant (Rhodasurf CET-2) in the experimental formulation, to improve hydrophilicity of material, resulted in a reduced percentage of elastic recovery of material, however, the changes were statistically not significant (Table 2).

## Strain-in-compression

Figure 2 and Table 2 reveal the strain-in-compression for the tested VPS impression materials. Experimental VPS had significantly higher ($p < 0.05$) strain-in-compressions values compared to the commercial VPS. Exp-V exhibited significantly the highest (Tukey's HSD test) strain-in-compression (7.08% ± 0.22%) while Elt M had the lowest values (3.15% ± 0.18%). Among commercial materials, no significant difference between Aq M and Extr M was noted. However, it was noted that the addition of a novel crosslinking agent *i.e.,* TFDMSOS (Exp II), significantly increased the percentage strain-in-compression values compared to formulations based on conventional cross-linking agents (Exp-I, Aq M, Elt M, Extr M). Also, it was observed that experimental formulations incorporating non-ionic surfactant (Rhodasurf CET-2) led to a further significant increase in strain-in-compression values and this effect was concentration dependant.

Figure 3 and Table 2 show the comparison between elastic recovery and strain-in-compression for all commercial and experimental VPS impression materials evaluated in this study. Among the experimental materials, there is a correlation between elastic recovery and strain-in-compression. With the addition of TFDMSOS in Exp-II the elastic recovery and strain-in-compression increase significantly compared to Exp-I (control). However, there is a negative correlation seen after addition of Rhodasurf CET-2 (non-ionic surfactant) in Exp III. With the addition of surfactant the elastic recovery is decreased while stain-in-compression is increased. It can also be noticed that with the increase in % amount of surfactant there is a consistent and significant decrease in elastic recovery and significant incurease in strain-in-compression in Exp-IV and Exp-V. Among commercial materials, no significant difference was seen.

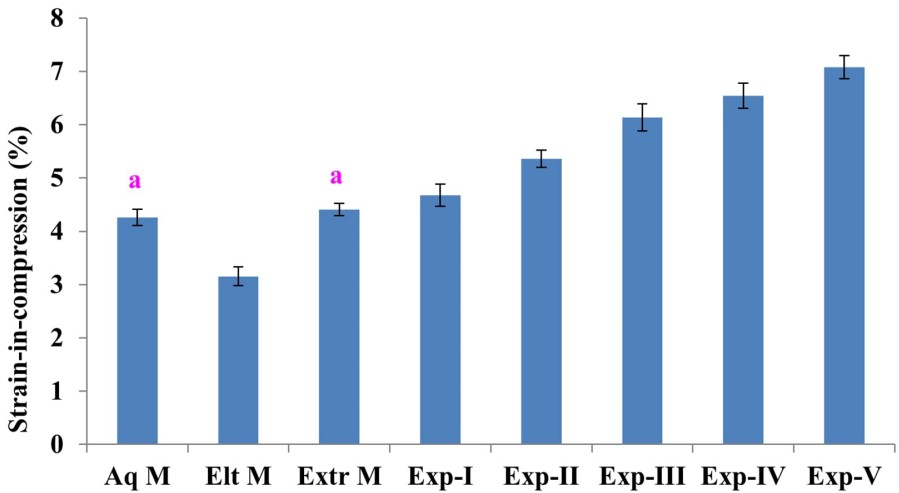

**Figure 2** Mean (±standard errors; $n = 12$) strain-in-compression of commercial and experimental polymeric VPS immediately after setting. Similar lowercase letters indicate no significant difference ($p > 0.05$).

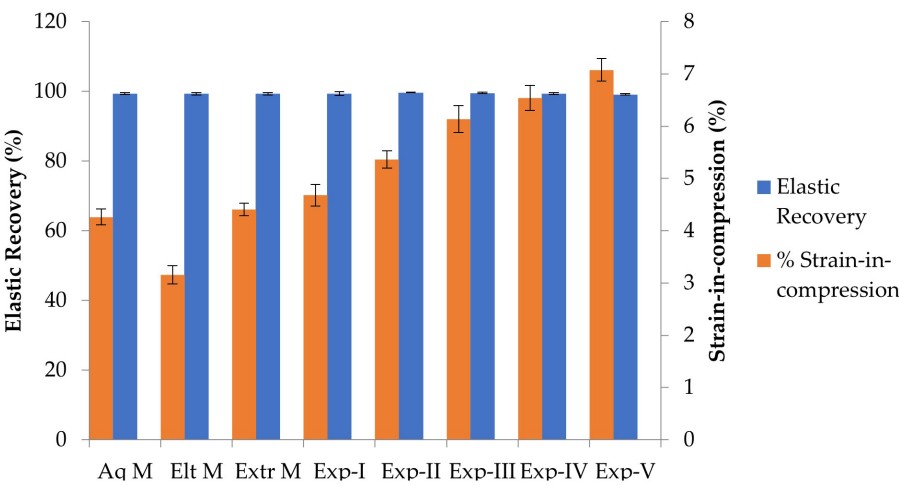

**Figure 3** Mean (±standard errors; $n = 12$) comparison of elastic recovery and strain-in-compression of commercial and experimental VPS immediately after setting.

## DISCUSSION

The elastic recovery of the impression materials plays a major role in the accurate reproduction of the oral cavity. The ability of elastomeric impression materials to revert to their actual form upon removal of the applied stress is related to their coiled wrapped and kinked molecular chains. These polymer chains stretch in the direction of stress and, recoil back on releasing the force, gaining their original shape and form (*Hamalian, Nasr & Chidiac, 2011*; *Klooster, Logan & Tjan, 1991*). In the present study, elastic recovery and

strain-in-compression of commercial and experimental VPS impression materials were compared.

Values for elastic recovery for commercial and experimental VPS impression materials ranged from 99.11 to 99.64%. These values were within the range set by the International Standards Organization (*ISO4823, 2007*) which requires ≥96.5%. Similar results were reported by *Lawson, Burgess & Litaker (2008)*, who noted that elastic recovery of five tested VPS (Aquasil Ultra, Examix, Genie, Imprint 3, and StandOut) and one hybrid impression material (Senn) ranges from 99.34 to 99.83%. In another study, *Lu, Nguyen & Powers (2004a)* investigated the elastic recovery of two VPS (Flexitime and Imprint II) and one polyether (Impregum). It was noted that Flexitime, Imprint II and Impregum had 99.60, 99.75 and 99.19% elastic recoveries respectively.

Similar percentage elastic recovery was noted for the commercial materials (Aq M: 99.32%, Elt M: 99.31%, Extr M: 99.27%) and Exp-I (99.32%) containing conventional cross-linking agent polymethylhydrosiloxane. However, on incorporating a novel cross-linking agent (TFDMSOS) in Exp-II, an increase in elastic recovery (99.65%) was observed which was statistically significant. The greater elastic recovery of Exp-II is attributed to excellent crosslinking of TFDMSOS with functional groups of vinyl-terminated poly (dimethylsiloxane) pre-polymer as each molecule of TFDMSOS can bond to four functional groups of pre-polymer (*Din et al., 2018a*). Similar results have been reported in the literature indicating the amount of permeant deformation of an impression material is greatly influenced by the degree of cross-linking of the polymeric chains (*Singer et al., 2022*). The degree of polymerization also affects other mechanical properties of elastomeric impression materials such as tear strength and % elongation-at-break (*Din et al., 2018a*; *Din et al., 2018b*). It is reported that both the properties improved due to increased cross-linking of polymeric chains.

The strain-in-compression was also calculated to assess the rigidity of impression materials so that it can be removed from the mouth or cast without permanent deformation after setting, and to resist deformation when the dental stone is poured. All tested impression materials have values for % strain-in-compression within the *ISO4823 (2007)* limits. Experimental VPS impression materials had higher strain-in-compression values indicating improved flexibility of experimental material (*Lu, Nguyen & Powers, 2004b*). Therefore, a positive correlation between elastic recovery and strain-in-compression was noted (Fig. 3 and Table 2). Additionally, it was observed that the incorporation of the wetting agent (Rhodasurf CET-2), further significantly increased strain-in-compression values. This was contradictory to the results of *Lu, Nguyen & Powers (2004b)* who noticed that flexible materials have less cross-linking and have better elastic recovery. This might be due to the difference in the composition of the materials used in the present study.

The ability to undergo greater elastic recovery is a desirable property of impression materials as it ensures an accurate impression which in turn guarantees a correct fit of the prosthesis. The experimental VPS impression materials in this study show greater elastic recovery than their commercial counterparts. Additionally, in previous studies, same experimental material has proven to have improved wettability, percentage elongation,

tear strength and minimal distortion after disinfection making them a much more suitable option for impression taking (*Din et al., 2018a*; *Din et al., 2018b*; *Din et al., 2021*).

A limitation of this study is that it is conducted in an *in-vitro* environment under laboratory conditions. To strengthen the claim of experimental VPS as a superior impression material, it is necessary to conduct further research in intra-oral, *in-vivo*, conditions.

## CONCLUSIONS

The addition of a novel cross-linking agent (TFDMSOS) showed improved elastic recovery and strain-in-compression, while the addition of a non-ionic surfactant also significantly increased strain-in-compressions values for all experimental VPS. All tested materials comply with ISO standards. In the future, biocompatibility testing followed by clinical trials should be undertaken, and material selection should be based on adequate knowledge of the properties of materials to improve clinical success.

### Funding
The authors received no funding for this work.

### Competing Interests
The authors declare there are no competing interests.

### Author Contributions
- Shahab Ud Din conceived and designed the experiments, performed the experiments, prepared figures and/or tables, and approved the final draft.
- Osama Khattak conceived and designed the experiments, prepared figures and/or tables, and approved the final draft.
- Farooq Ahmad Chaudhary conceived and designed the experiments, performed the experiments, prepared figures and/or tables, and approved the final draft.
- Asfia Saeed conceived and designed the experiments, performed the experiments, authored or reviewed drafts of the article, and approved the final draft.
- Azhar Iqbal performed the experiments, analyzed the data, prepared figures and/or tables, and approved the final draft.
- Jamaluddin Syed performed the experiments, analyzed the data, authored or reviewed drafts of the article, and approved the final draft.
- Alaa Ahmed Kensara analyzed the data, authored or reviewed drafts of the article, and approved the final draft.
- Thani Alsharari analyzed the data, authored or reviewed drafts of the article, and approved the final draft.
- Mohammed Mustafa analyzed the data, authored or reviewed drafts of the article, and approved the final draft.

- Sherif Elsayed Sultan analyzed the data, prepared figures and/or tables, and approved the final draft.
- Mangala Patel performed the experiments, prepared figures and/or tables, authored or reviewed drafts of the article, and approved the final draft.

## Data Availability

The raw data is available in the Supplementary Files.

## Supplemental Information

Supplemental information for this article can be found online at http://dx.doi.org/10.7717/peerj.15677#supplemental-information.

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
