# Peer review of "Comparison of the elastic recovery and strain-in-compression of commercial and novel vinyl polysiloxane impression materials incorporating a novel crosslinking agent and a surfactant"

_PeerJ, doi:10.7717/peerj.15677_

## Round 0.1 · original submission · Major Revisions

Please address the concerns of all reviewers and amend the manuscript accordingly.

·

Basic reporting

The topic of the article is interesting and sheds light on developing newer elastomeric impression material with better properties.

Experimental design

Information regarding measurement devices and surfactants incorporated in the experimental group should be mentioned.

Validity of the findings

The findings are valid.

Additional comments

Corrections are mentioned in the pdf file.

Reviewer 2 ·

Basic reporting

There is a lack of scientific writing. Eight plus authors from different regions of the world, but the quality of English not reflecting professional writing. Look like an attachment of statements from the published work.

There is a need to expand the literature information in the introduction and discussion part with some latest papers published in similar areas and compare why this work concludes the clinical trials. Is this work concluded all the conditions for bringing the clinical trials? If yes, provide biocompatibility evidence in your experiment.

Authors hypotheses are not justifying the result outcomes. Justify this hypothesis well in your result and discussion part.

Experimental design

Experimental work is weak. Authors did not discuss why they used this composition (Table-1) of the experimental group and how they homogeneously blended all ingredients.

There are many mistakes in presenting the method part. Authors need to revise scientific writing from senior authors. They can improve it again.

This work will not go to fill the gap in clinical dentistry. It's just a hypothesis; they have done multiple experiments with different compositions. But they not defined the chemistry behind it.

Validity of the findings

Researchers just checked the two properties of the experimental samples. Which is not enough to publish this work. They can add more about biocompatibility, and how its pouring time. flow of the material and how they provide the replica.

These all guidance can expand the work and novelty.

Conclusion need revision and add some lines regarding the limitation of the work before conclusion.

Reviewer 3 ·

Basic reporting

Please focus on " what is the need of doing this investigation?'' " what are the current loop holes in the scientific evidence available?" specially in the introduction section.

Experimental design

no comment

Validity of the findings

The novelty if any please mention in the introduction section, please specify the rationale of the study, in last paragraph

Additional comments

Methods: i would like to see the study site, ethical clearance if any, the laboratory where testing were carried out. The authors are requested to add study design. if your study is a trial of in vitro category. please provide the trial registration organization details and number. add the check list of CONSORT guidelines as a supplementary file.

in discussion add future recommendation to the fellow researchers or directions of further work in this area.

---

## Round 0.2 · Minor Revisions

Please address the remaining concerns of the reviewer and amend the manuscript accordingly.

I think that since the reviewer has these concerns, something was not clear enough in the manuscript and there is a chance that other readers will have similar queries. Therefore, in my view, a brief statement providing these clarifications should be added to the manuscript.

Reviewer 3 ·

Basic reporting

see the comment

Experimental design

comment provided to the author...not answered in first round

Validity of the findings

no comment

Additional comments

Comment 4: Additional comments: Methods: I would like to see the study site, ethical clearance if any, and the laboratory where testing was carried out. The authors are requested to add a study design. if your study is a trial of in vitro category. please provide the trial registration organization details and number. add the checklist of CONSORT guidelines as a supplementary file. in discussion add future recommendations to fellow researchers or directions for further work in this area.

Response: This study was carried out in Institute of Dentistry, Floor 2, Francis Bancroft Building, Mile End Campus, Queen Mary University of London, UK. No ethical clearance and trial registration was required for this study. Future directions and recommendations have added for future studies and research as advised.

Comment.

The authors did not respond to the comment in the manuscript file. Why was the trial not registered in a (WHO trial registry)? It is a standard protocol as per editorial guidelines. Where is the response on CONSORT guidelines or the nature of the study? How come the project was not approved by an authority in a university like Queen Marry prior to conducting the research?

---

## Round 0.3 · accepted · Accept

All remaining issues were addressed and therefore the revised manuscript is acceptable now.